# Characterizing the actin-binding ability of Zasp52 and its contribution to myofibril assembly

**Kuo An Liao**, **Nicanor González-Morales**, **Frieder Schöck** *

Department of Biology, McGill University, Montreal, Quebec, Canada

These authors contributed equally to this work.
* frieder.schoeck@mcgill.ca

**Data Availability Statement:** All relevant data are within the paper and its Supporting Information files.

**Funding:** This work was supported by operating grant MOP-142475 and PJT-155995 to F.S. from

## Abstract

In sarcomeres, α-actinin crosslinks thin filaments and anchors them at the Z-disc. *Drosophila melanogaster* Zasp52 also localizes at Z-discs and interacts with α-actinin via its extended PDZ domain, thereby contributing to myofibril assembly and maintenance, yet the detailed mechanism of Zasp52 function is unknown. Here we show a strong genetic interaction between actin and Zasp52 during indirect flight muscle assembly, indicating that this interaction plays a critical role during myofibril assembly. Our results suggest that Zasp52 contains an actin-binding site, which includes the extended PDZ domain and the ZM region. Zasp52 binds with micromolar affinity to monomeric actin. A co-sedimentation assay indicates that Zasp52 can also bind to F-actin. Finally, we use in vivo rescue assays of myofibril assembly to show that the α-actinin-binding domain of Zasp52 is not sufficient for a full rescue of *Zasp52* mutants suggesting additional contributions of Zasp52 actin-binding to myofibril assembly.

## Introduction

Striated muscles, including skeletal and cardiac muscle, contain highly organized myofibrils, composed of repeating functional elements called sarcomeres. In a sarcomere, the smallest contractile unit of muscle, myosin thick filaments, which are anchored at M-lines, and actin thin filaments, which are attached to Z-discs, cooperate to mediate muscle contraction. The Z-disc defines the lateral boundary of the sarcomere and contains multi-protein complexes essential for the maintenance of muscle structural integrity, tension transmission, and signal transduction [1–5]. A crucial component of Z-discs is α-actinin, which crosslinks and organizes actin filaments at the Z-disc. In addition, members of the Alp/Enigma protein family have recently been characterized as Z-disc proteins and have been shown to play important roles in Z-disc maintenance and myofibril assembly [6, 7]. In vertebrates, the Alp/Enigma family members consist of ZASP/Cypher/Oracle/LDB3/PDLIM6, ENH/PDLIM5, PDLIM7/ENIGMA/LMP-1, CLP36/PDLIM1/Elfin/hCLIM1, PDLIM2/Mystique/SLIM, ALP/PDLIM3 and RIL/PDLIM4. The Enigma family includes the first three members ZASP, ENH and PDLIM7. These subfamily members contain one N-terminal PDZ domain and three LIM

the Canadian Institutes of Health Research (https://cihr-irsc.gc.ca). The funders had no role in study design, data collection and analysis, decision to publish, or preparation of the manuscript.

**Competing interests:** The authors have declared that no competing interests exist.

domains at the C-terminus. The next four members have been classified into the ALP subfamily and have one N-terminal PDZ domain and only one C-terminal LIM domain [8]. In *Drosophila*, Zasp52 is the canonical member of the Zasp PDZ domain family, containing a PDZ domain, a Zasp-like motif (ZM) and four LIM domains. Zasp66 and Zasp67 are paralogs of Zasp52 in *Drosophila* and other insects, but they only feature the N-terminal PDZ domain and a weakly conserved ZM motif [9]. Many different Zasp52 splice isoforms have been identified resulting in many different proteins, some of which are restricted to specific muscle types [10, 11]. Our group and other authors demonstrated that Zasp52 colocalizes with α-actinin at Z-discs and is required for both initial sarcomere assembly and sarcomere maintenance [12–14]. Furthermore, Zasp52, Zasp66 and Zasp67 cooperate and function partially redundantly in Z-disc formation and myofibril assembly [6], but also carry out some unique functions [9]. Zasp52 biochemically interacts with α-actinin via an extended PDZ domain, and the PDZ domain is required for myofibril assembly [7]. On the other hand, the LIM and ZM domains play a crucial role in mediating self-interaction leading to Z-disc growth or aggregate formation [15]. Mutations of Zasp52 orthologs across species cause similar muscle defects. Mutants in the single *C. elegans* ortholog *alp-1* show actin myofilament disorganization during times of increased muscle load [16, 17]. In vertebrates, knockdown of *cypher* leads to deformation of somites and improper heart development in zebrafish [18]. Similar to zebrafish, Cypher null mutant mice display disorganized and fragmented Z-discs and exhibit dilated cardiomyopathy [8, 19, 20]. Mutations in the human ortholog ZASP have been identified in different forms of myofibrillar myopathies and cardiomyopathies [21–23]. These studies demonstrate the importance of Zasp52 and its orthologs in muscle biology. Biochemical assays identified the internal ZM motif-containing region of human ZASP as an actin-binding domain [24, 25], providing an additional explanation for the central role of Zasp proteins as scaffolding proteins at Z-discs. In this study, we therefore characterized the actin-binding ability of *Drosophila* Zasp52 and its importance for myofibril assembly.

## Results and discussion

### *Zasp52* genetically interacts with *actin* causing severe myofibril defects

Yeast-two hybrid and *in vitro* binding assays showed that human ZASP binds actin [24]. We therefore wanted to know if the *Drosophila* ortholog Zasp52 and actin work together *in vivo* during myofibril assembly. To this end, we investigated the genetic interaction between *Zasp52* and *Act88F*. *Zasp52^MI02988^* heterozygotes are indistinguishable from wild type (Fig 1A, 1B and 1E). *Act88F* is one of six actin-coding genes in *Drosophila* and is the major and possibly the only one expressed in indirect flight muscles [26]. *Act88F^KM88^* null mutants are flightless and lack all thin filaments [27]. *Act88F^KM88^* heterozygous flies are haploinsufficient and exhibit some myofibril fraying and loss of sarcomere structure compared to wild type (Fig 1C and 1E). Intriguingly, transheterozygous *Zasp52^MI02988^*/+; *Act88F^KM88^*/+ flies feature a major disruption of Z-discs and myofibrils with many fragmented Z-discs or completely unrecognizable sarcomere structure (Fig 1D and 1E). These results show that Zasp52 interacts genetically with actin, and this interaction plays a critical role in Z-disc and myofibril assembly.

### Zasp52 is an actin-binding protein

Next we wanted to know if Zasp52 binds biochemically to actin. We tested two GST constructs covering full length Zasp52, Zasp52-PK and Zasp52-LIM234, with biotinylated G-actin by *in vitro* GST pull down assays. Zasp52-PK robustly binds actin, while Zasp52-LIM234 and GST alone do not (Fig 2, Fig 3A and S1 Fig).

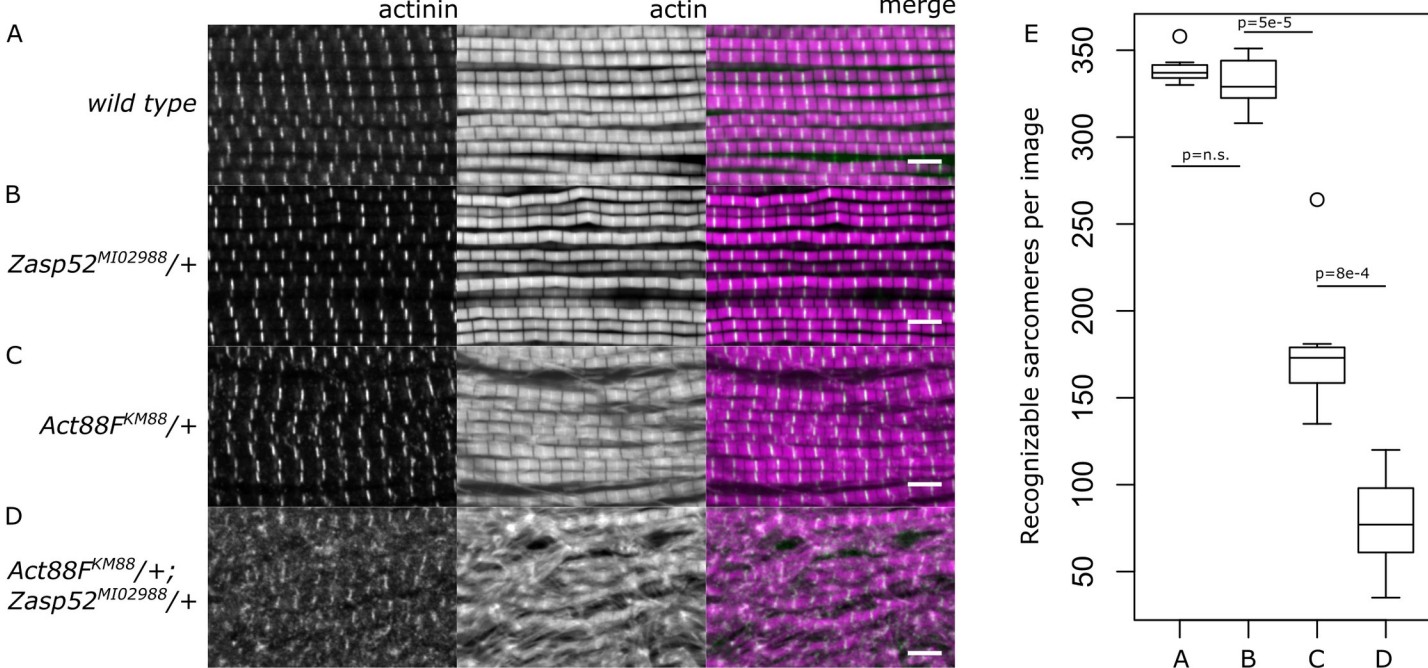

**Fig 1. *Zasp52^MI02988* and *Act88F^KM88* interact genetically during myofibril assembly.** Confocal microscopy images of IFM of wild type flies and heterozygotes stained with phalloidin (magenta) to visualize myofibrils, and anti-α-actinin antibody (green) to visualize Z-discs. (**A**) Wild type myofibrils show no defects with properly formed sarcomeres. (**B**) *Zasp52^MI02988*/+ heterozygotes look indistinguishable from wild type. (**C**) In the *Act88F^KM88*/+ heterozygote, myofibrils form properly, but some frayed myofibrils and loss of sarcomere structure occurs compared to wild type flies. (**D**) The *Zasp52^MI02988*/+; *Act88F^KM88*/+ transheterozygotes frequently exhibit frayed myofibrils and fragmented Z-discs, and sometimes complete loss of sarcomere integrity. (**E**) Box plot of quantification of remaining sarcomeres per image in wild type flies, and various transheterozygous mutants. n = 7 muscle fibers. Scale bar, 5 μm. P-values were calculated using Welch's two-sample t-test followed by Bonferroni correction.

To confirm the direct binding of Zasp52-PK to actin and obtain the binding affinity of this interaction, we used surface plasmon resonance imaging, a method previously used to determine the dissociation constant of actin-binding proteins to G-actin [28]. We tethered purified GST-Zasp52-PK to the biosensor chip and flowed different concentrations of G-actin over the

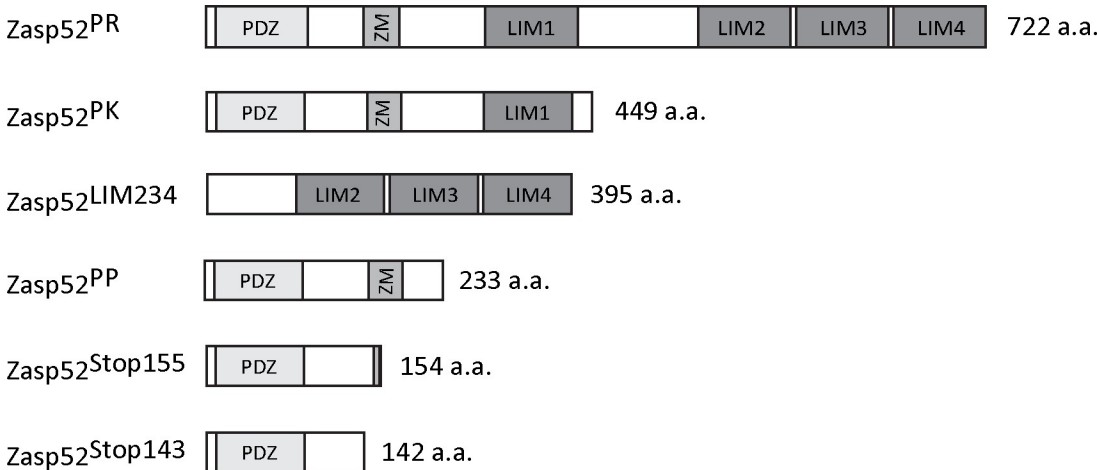

**Fig 2. Schematic of different Zasp52 proteins used.** Size of proteins is indicated in amino acids (a.a.). The full-length protein Zasp52-PR is shown for comparison.

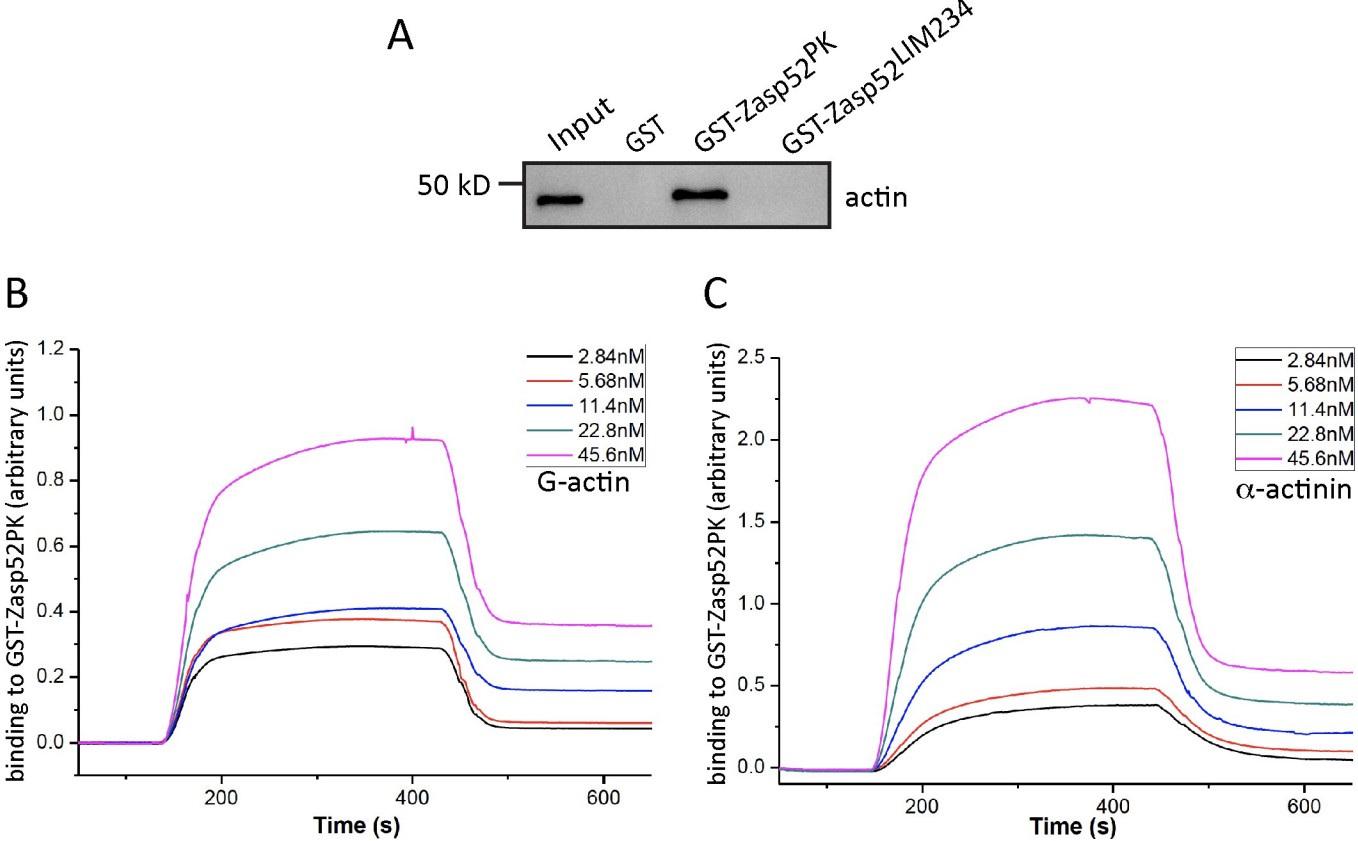

**Fig 3. Zasp52 binds actin with micromolar affinity.** (**A**) GST pull-down assay. Zasp52-PK, the N-terminal half of Zasp52, binds actin, while Zasp52-LIM234, the C-terminal half of Zasp52, and GST alone do not bind. Binding results were observed in at least three independent experiments. (**B, C**) Surface plasmon resonance imaging of the real-time binding to GST-Zasp52-PK tethered to the biosensor chip. (**B**) Binding of indicated concentrations of G-actin flown into the chamber at 100 s and replaced with buffer at 500 s. (**C**) Binding of indicated concentrations of α-actinin flown into the chamber at 100 s and replaced with buffer at 500 s. Binding is measured in real time in arbitrary units.

chip to monitor binding in real time (Fig 3B). From the association and dissociation curves we calculated a dissociation constant ($K_d$) of 1.1 x $10^{-6}$ M. As a control, we also measured the binding affinity to α-actinin (Fig 3C), resulting in a $K_d$ of 8.65 x $10^{-8}$ M. The higher binding affinity of α-actinin to Zasp52 is consistent with our qualitative observations from pull-down assays. In summary, our results indicate that Zasp52 interacts directly with G-actin with an affinity in the micromolar range.

## Zasp52 binds filamentous actin

In order to affect thin filament structure at the Z-disc, Zasp52 ought to bind filamentous actin (F-actin). We therefore purified a 6xHis- and Flag-tagged Zasp52-PK (His-Zasp52-PK-Flag) to avoid GST-induced aggregation. Binding to F-actin was examined by high-speed co-sedimentation assays. In the absence of F-actin, His-Zasp52-PK-Flag stays in the supernatant after high-speed centrifugation (Fig 4A, right side). In contrast, F-actin on its own sediments into the pellet at high-speed centrifugation (Fig 4A, left side). When Zasp52 is incubated with F-actin, a small fraction of His-Zasp52-PK-Flag accumulates in the pellet together with F-actin (Fig 4A, middle), whereas the control protein BSA always stays in the supernatant (Fig 4B), indicating that Zasp52 can also bind F-actin.

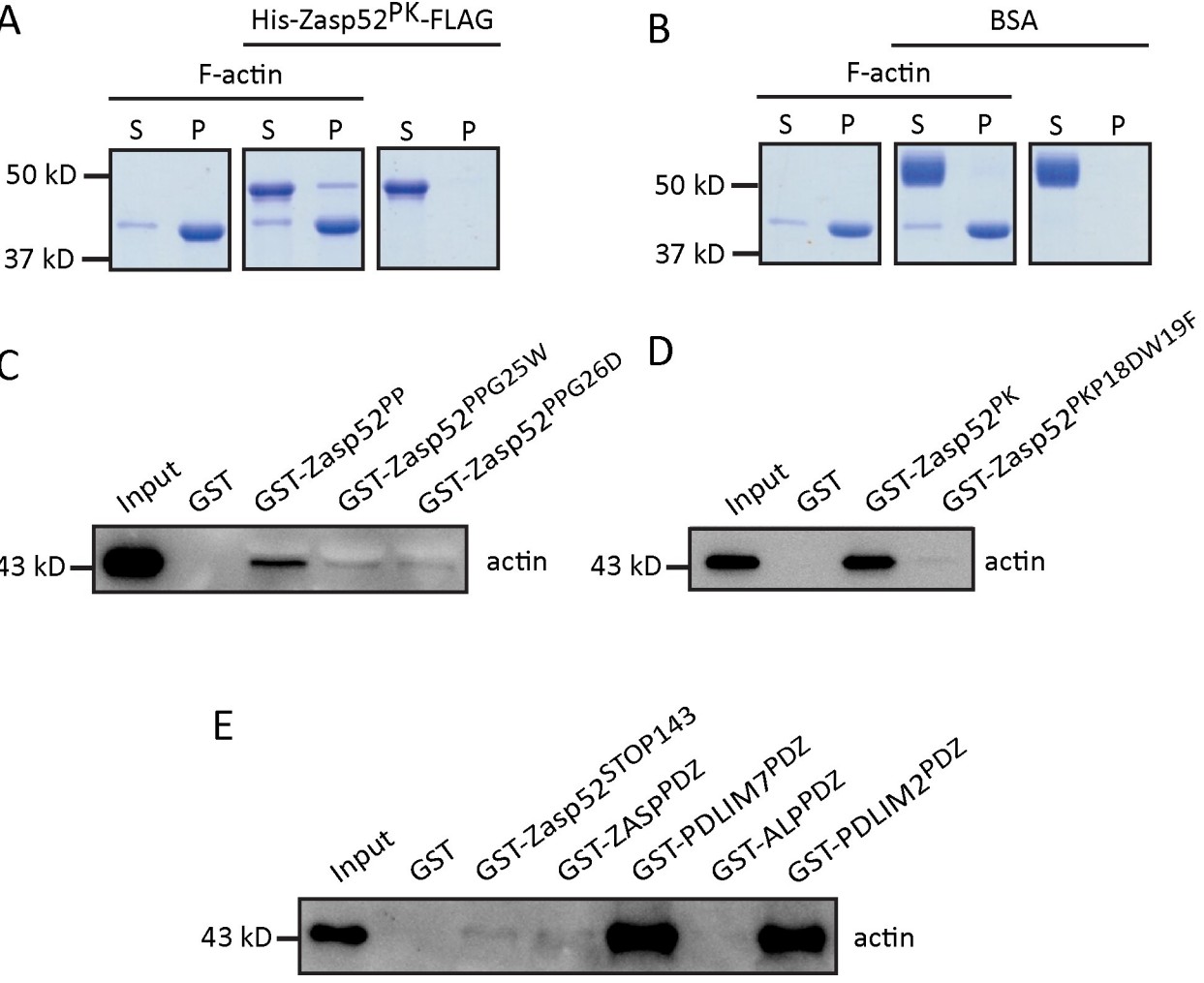

**Fig 4. Zasp52 binds filamentous actin and the extended PDZ domain of Alp/Enigma proteins contributes to actin binding.** (**A**, **B**) Coomassie blue-stained SDS-PAGE gel showing the results of high-speed co-sedimentation assay of purified His$_6$-Zasp52-PK-FLAG or pure BSA and F-actin. S and P indicate the supernatant and the pellet after high-speed centrifugation, respectively. (**A**) A small amount of His$_6$-Zasp52-PK-FLAG precipitates together with F-actin after high-speed centrifugation. (**B**) The control protein BSA remains in the supernatant. (**C**, **D**) GST pull-down assays with PWGFRL motif point mutations. (**C**) G25W and G26D in Zasp52-PP reduce actin binding. (**D**) P18DW19F in Zasp52-PK strongly reduces actin binding. (**E**) GST pull-down assay with representative purified extended PDZ domains of *Drosophila* Zasp52, and human ZASP, PDLIM7, ALP, and PDLIM2. We observe strong actin binding of PDLIM7 and PDLIM2, and weak or no binding of the other extended PDZ domain proteins. Binding and co-sedimentation results were observed in at least three independent experiments.

## In Zasp52 the extended PDZ domain and ZM region is required for actin binding

Human ZASP binds actin with the ZM region [24]. We therefore first tested the shortest Zasp52 isoform, the 233 amino acid long Zasp52-PP, which contains only the PDZ domain and the ZM region, for actin binding. In GST pulldown assays, Zasp52-PP can still bind to actin, indicating that the extended PDZ domain and ZM region is sufficient for binding to actin (Fig 2, Fig 4C and S1 Fig). Next we wanted to determine if the PDZ domain is also involved in actin binding. We therefore tested two point mutations in the PWGFRL motif of the PDZ domain (G25W and G26D). Both mutants considerably reduce binding to actin compared to Zasp52-PP, but do not completely abolish it (Fig 2, Fig 4C and S1 Fig). We also tested a double mutant in the PDZ domain (P18D W19F) of the larger Zasp52-PK protein, which

binds actin as well as Zasp52-PP. This mutant also strongly disrupts actin binding (Fig 2, Fig 4D and S1 Fig). These data imply that both the ZM and PDZ domains are essential contributors to the actin-binding capacity of Zasp52, suggesting a more complex molecular basis for this interaction than previously recognized.

The Alp/Enigma family has seven members in vertebrates. To assess the actin-binding ability of this protein family, we purified the extended PDZ domains of representative members and tested their binding to actin. The extended PDZ domain of ALP showed no binding, the extended PDZ domain of ZASP showed weak binding like Zasp52, but the extended PDZ domains of PDLIM2 and PDLIM7 showed robust binding to actin (Fig 2, Fig 4E and S1 Fig). This indicates that extended PDZ domains of Alp/Enigma proteins can bind actin, although highly variably.

## α-actinin-binding of Zasp52 alone is not enough to restore Zasp function

We have previously described *Zasp52$^{MI02988}$*, a mutation in the N-terminal region of Zasp52 that stops the translation of isoforms that contain the PDZ domain [7]. The effect of this mutant on myofibril assembly can be fully restored by the expression of the Zasp52-PK isoform. To further investigate this mutation, we tested the rescuing ability of two smaller variants. These variants are expressed at similar levels and both localize to Z-discs [15]. Interestingly, Zasp52-PP, the smallest isoform that contains PDZ and ZM domains, can also fully rescue *Zasp52$^{MI02988}$*/Df myofibril defects (Fig 5A–5C and 5E). Zasp52-PP binds both α-actinin and actin similarly to Zasp52-PK (Fig 4) [7]. In contrast, the truncated form Zasp52-STOP143 lacks the ZM region, which may reduce its binding to actin, but still strongly binds α-actinin [7]. Zasp52-STOP143 can only partially rescue myofibril assembly. A significant proportion of frayed myofibrils are still present, indicating that α-actinin binding alone is not sufficient to restore Zasp function (Fig 5D and 5E)

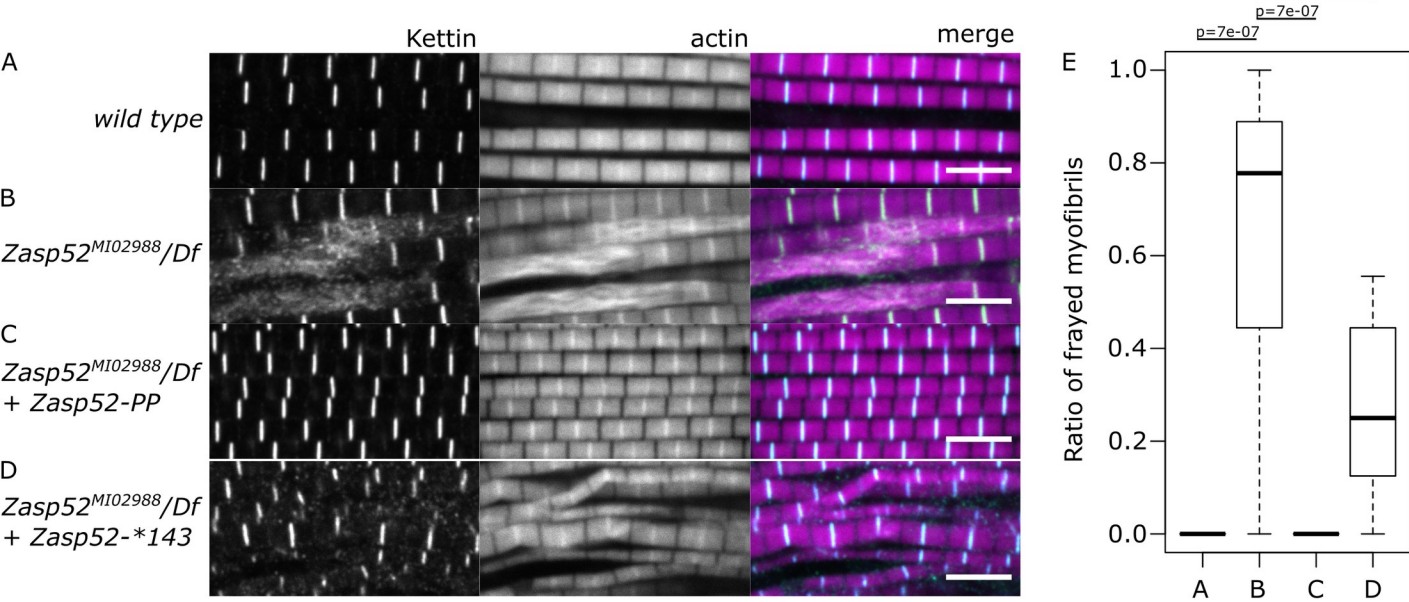

**Fig 5. Limited rescue ability of a Zasp52 protein lacking the ZM region.** Confocal microscopy images of IFM of different Zasp52 genetic backgrounds stained with phalloidin (magenta) to visualize myofibrils, and anti-kettin antibody (green) to visualize Z-discs. (**A**) Wild type myofibrils show no defects with properly formed sarcomeres. (**B**) In the *Zasp52$^{MI02988}$*/Df, myofibrils appear frayed and the Z-disc does not form. (**C**) In *Zasp52$^{MI02988}$*/Df rescued by Zasp52-PP flies, myofibrils look indistinguishable from wild type. (**D**) In *Zasp52$^{MI02988}$*/Df rescued by Zasp-*143 muscles, myofibrils are frequently frayed and smaller. (**E**) Box plot quantification of the ratio of frayed myofibrils in different *Zasp52* genetic backgrounds. UH3-Gal4 was used for the expression of Zasp52 transgenes. n = 10 muscle fibers. Scale bar, 5 μm. P-values were calculated using Welch's two-sample t-test followed by Bonferroni correction.

In this report we first examined whether the physical interaction of Zasp52 with actin is important for actin filament organization *in vivo*, in particular during myofibril assembly. We observe a genetic interaction of actin and *Zasp52*, leading to severe myofibril defects in trans-heterozygotes (Fig 1). This suggests that the physical interaction of Zasp52 with actin filaments plays a critical role in myofibril assembly.

Zasp52 and its orthologs mouse Cypher and human ZASP are critical Z-disc proteins functioning in Z-disc formation and myofibril assembly. They interact directly with α-actinin, a core structural protein crosslinking actin thin filaments at Z-discs [7, 20, 29]. The C-terminal LIM domains of Cypher and ZASP have been shown to bind protein kinase C and the β1 integrin tail, respectively [30, 31]. The function of the ZM motif and its interacting proteins is less well defined. A previous study showed that the ZM region of ZASP colocalizes with α-actinin but does not directly interact with α-actinin [32]. Later it was shown that the ZM region of ZASP is responsible for the direct interaction with skeletal muscle α-actin 1, but mutations in the ZM region that cause myopathies could not be shown to affect actin binding [24]. We recently showed that the ZM domain is crucial for mediating self-interaction of Zasp proteins [15]. Here we show that in Zasp52 the PDZ domain and the ZM region is additionally required for optimal actin binding. We used surface plasmon resonance measurements and determined a micromolar binding affinity to monomeric actin, similar to results for other G-actin binding proteins [28]. We also measured the binding affinity of Zasp52 to α-actinin. As expected, Zasp52 binds better to α-actinin than to actin. The dissociation constant is very similar to that of human ALP binding to α-actinin [33, 34].

To test the ability of Zasp52 to bind filamentous actin, we performed a co-sedimentation assay showing Zasp52-PK co-sediments with F-actin (Fig 4A). Similar results have been obtained for human ZASP [24].

Intriguingly, site-directed mutagenesis experiments suggest that the PDZ domain also contributes to actin binding (Fig 4C and 4D). We have only recently shown the importance of a C-terminal extension of the PDZ domain for the interaction of Zasp52 with α-actinin [7], suggesting a similar mechanism applies to actin binding. The P18DW19F mutation in Zasp52-PK, and G25W and G26D in Zasp52-PP, all within a conserved part of the PDZ domain we call PWGFRL motif, strongly reduce the binding of Zasp52 to actin (Fig 4C and 4D). Furthermore, testing several members of the Alp/Enigma family revealed that some extended PDZ domains (PDLIM7 and PDLIM2) bind robustly to actin (Fig 4E). We propose that binding of the PDZ domain to actin was overlooked, because either only the PDZ domain proper was used, or an extended PDZ domain that only weakly interacts with actin without the ZM region. The affinity constant for actin binding of Zasp52 is considerably lower than that reported for ZASP [25], which could be due to the variability in actin binding that we observed qualitatively with different extended PDZ domains (Fig 4E), or could be explained by the sequence divergence between *Drosophila* and human Zasp proteins. We propose that in Zasp52 the extended PDZ domain plus the ZM region is required for optimal actin binding, whereas optimal α-actinin binding requires only the extended PDZ domain.

Our in vivo rescue assays demonstrate that α-actinin binding is not sufficient for Zasp52 to mediate myofibril assembly (Fig 5). *Zasp52*[MI02988] is a hypomorphic allele that leaves some LIM domain-containing isoforms intact, and also allows minimal expression of PDZ domain-containing isoforms [7]. It is viable, and therefore ideally suited to analyze the contribution of PDZ and ZM region to myofibril assembly. We have previously shown that Zasp52 without PDZ domain, that is, Zasp52 neither able to bind actin nor α-actinin, cannot rescue the mutant phenotype, while Zasp52-PK containing PDZ, ZM and LIM1 domain, can fully rescue [7]. Here we show that Zasp52-PP, the minimal isoform binding both actin and α-actinin, confers full rescue, whereas Zasp52-STOP143, which presumably binds mostly α-actinin, can only

partially rescue (Fig 5). This identifies α-actinin binding as crucial for the initial establishment of Z-discs. Unfortunately, the importance of actin binding cannot be unambiguously demonstrated, because the ZM region present in Zasp52-PP likely contributes to actin binding, but also mediates self-interaction [15]. We can currently not separate these functions through individual amino acid mutations. However, the genetic interaction of actin and Zasp52 (Fig 1) suggests that at least part of the phenotype observed with Zasp52-STOP143 may be due to reduced actin binding. Future work should address if Zasp52 can bind simultaneously to actin and α-actinin, as well as identify mutations that uniquely disrupt actin versus α-actinin binding.

Finally, our *in vitro* studies suggest that several Alp/Enigma proteins may contribute to actin binding, and in particular PDLIM7 and PDLIM2. In support of this notion, the Enigma family member PDLIM7 was shown to play a role in actin cytoskeletal organization, with loss of PDLIM7 abolishing the formation of stress fibers in mouse embryonic fibroblasts leading to platelet dysfunction [35]. Thus, the well-documented association of Alp/Enigma proteins with actin stress fibers in nonmuscle cells and thin filaments in muscle cells is likely due to both their actin as well as their α-actinin binding.

## Materials and methods

### Fly stocks and genetics

The following fly stocks were used: *GFP-Zasp52* (*Zasp52$^{G00189}$*; #6838); *Mef2-Gal4* (#27390), Df(2R)BSC427 (#24931), and *Zasp52$^{MI02988}$* (#41034) from the Bloomington *Drosophila* Stock Center; UAS-FSH-Zasp52-PK [7]; UAS-Flag-Zasp52-PP and UAS-Flag-Zasp52-Stop143 [15]; *Act88F$^{KM88}$* was a kind gift from John Sparrow. To test the genetic interaction of *Zasp52* with *Actin88F*, *Zasp52$^{MI02988}$*/CTG was crossed to *Act88F$^{KM88}$*/TM3, *Ser* or *y w* and incubated at 25˚C.

### Plasmids

GST-tagged constructs (Zasp52-PK, Zasp52-LIM234, Zasp52-PK$^{P18DW19F}$, Zasp52-PP, Zasp52-PP$^{G25W}$, Zasp52-PP$^{G26D}$) are previously described [6, 7]. Zasp52-PK-FLAG, ZASP-PDZ (amino acid 1–136 of human LDB3 isoform 2), PDLIM7-PDZ (amino acid 1–137 of human PDLIM7 isoform 1), ALP-PDZ (amino acid 1–136 of human ALP isoform 1), and PDLIM2-PDZ (amino acid 1–136 of human PDLIM2 isoform 3) were synthesized by GenScript. Zasp52-PK-FLAG was then cloned into pRSETA (ThermoFisher Scientific) to give His$_6$-Zasp52-PK-FLAG. All others were cloned into pGEX-5X-1 (GE Healthcare) to give GST fusions. All plasmids were sequenced to confirm that the coding regions were in-frame with the appropriate tag.

### Purification of Zasp52-PK

Recombinant 6xHis-tagged Zasp52-PK-FLAG was overexpressed in *E. coli* BL21 bacteria by standard procedures. Bacterial cells were harvested by centrifugation, and then lysed by sonication in 20 mM Tris-HCl pH 8, 200 mM NaCl, 10 mM imidazole, 1 mM DTT, 5% glycerol, 0.2% Triton X-100, 1 mg/ml lysozyme and complete EDTA-free protease inhibitor (Roche). The extract was clarified by centrifugation and 0.45 μm filtration, and was incubated with prewashed Ni-NTA agarose beads (Qiagen) for 3 hours at 4˚C. The beads retaining the recombinant protein was washed three times with washing buffer (20 mM Tris-HCl pH 8, 250 mM NaCl, 10 mM imidazole, 1 mM DTT, 5% glycerol, and 0.2% Triton X-100), and bound protein was eluted with elution buffer (20 mM Tris-HCl pH 8, 150 mM NaCl, 200 mM imidazole, 2

mM DTT, 5% glycerol, and 0.2% Triton X-100). The protein was then dialyzed overnight at 4°C against buffer containing 20 mM Tris-HCl pH 8, 100 mM NaCl, 1 mM DTT, 5% glycerol, and 0.02% Triton X-100.

## Pull-down assays

*E. coli* strain BL21 bacteria expressing GST-tagged recombinant proteins were lysed by sonication in binding buffer (20 mM Tris-HCl pH 8, 200 mM NaCl, 1 mM DTT, 5% glycerol, 0.2% Triton X-100 and complete EDTA-free protease inhibitor from Roche), with 1 mg/ml lysozyme. The clarified cell extracts after centrifugation were filtered with 0.45 μm filters and coupled to prewashed glutathione-agarose beads (Santa Cruz Biotechnology) for 3 hours at 4°C. The beads retaining the GST-tagged proteins were washed three times with binding buffer with 250 mM NaCl and 0.5% Triton X-100. Subsequently, biotinylated G-actin (46.5 nM, Cytoskeleton) was added and incubated in binding buffer with 50 mM NaCl and 0.2 mM ATP for another 3 hours at 4°C. Final washes were in binding buffer with 175 mM NaCl, and the eluates were analyzed by SDS-PAGE and immunoblotting. The immunoreaction was visualized by ECL (Millipore). To detect biotinylated G-actin, the blot was probed with HRP-conjugated streptavidin (1:5000; ThermoFisher Scientific), and then detected by ECL (Millipore).

## Surface plasmon resonance binding affinity measurements

GST-Zasp52-PK was purified in two steps to a purity of about 71% with the AKTA avant 25 chromatography system (GE Healthcare). Various concentrations of GST-Zasp52-PK dissolved in buffer were manually printed onto the gold-coated (thickness 47 nm) PlexArray Nanocapture Sensor Chip (Plexera Bioscience) at 40% humidity. Each concentration was printed in replicate, and each spot contained 0.2 mL of sample solution. The chip was incubated in 80% humidity at 4°C for overnight, and rinsed with 10x PBST for 10 min, 1x PBST for 10 min, and deionized water twice for 10 min. The chip was then blocked with 5% (w/v) non-fat milk in water overnight, and washed with 10x PBST for 10 min, 1x PBST for 10 min, and deionized water twice for 10 min before being dried under a stream of nitrogen prior to use. SPRi measurements were performed with PlexArray HT (Plexera Bioscience). Collimated light (660 nm) passing through the coupling prism, was reflected off the SPR-active gold surface, and received by the CCD camera. Buffers and samples (G-actin or α-actinin from Cytoskeleton: AKL-99 and AT-01) were injected by a non-pulsatile piston pump into the 30 mL flow cell. Each measurement cycle contained four steps: washing with PBST running buffer at a constant rate of 2 mL/s to obtain a stable baseline, sample injection at 5 mL/s for binding, surface washing with PBST at 2 mL/s for 300 s, and regeneration with 0.5% (v/v) $H_3PO_4$ at 2 mL/s for 300 s. All the measurements were performed at 25°C. The signal changes after binding and washing (in arbitrary units) are recorded as the assay value. Selected protein-grafted regions in the SPR images were analyzed, and the average reflectivity variations of the chosen areas were plotted as a function of time. Real-time binding signals were recorded and analyzed by Data Analysis Module software (Plexera Bioscience). Kinetic analysis was performed using BIAevaluation 4.1 software (Biacore). Purification and affinity measurements were performed by Creative Biolabs.

## Actin co-sedimentation assay

Rabbit skeletal muscle G-actin (15 μM, Cytoskeleton) was polymerized in polymerization buffer (5 mM Tris-HCl pH 7.5, 40 mM KCl, 2 mM $MgCl_2$, 1 mM ATP, 1 mM DTT, and 0.1% Triton X-100) for 1 hour at room temperature. The reaction mixtures were centrifuged at 12,000 × g for 20 minutes, and the supernatant was incubated with purified $His_6$-

Zasp52-PK-FLAG (7 μM) or pure BSA (7 μM, BioShop) for 1 hour at room temperature. Subsequently, the samples were centrifuged at 150,000 × g for 1 hour to pellet F-actin and F-actin binding proteins for high-speed co-sedimentation experiments. Comparable amounts of supernatants and resuspended pellets were analyzed by SDS-PAGE and Coomassie Brilliant Blue staining.

### Immunofluorescence of indirect flight muscles

Half thoraces were glycerinated (20 mM Na-Phosphate pH 7.2, 2 mM MgCl$_2$, 2 mM EGTA, 5 mM DTT, 0.5% Triton X-100, 50% glycerol) overnight at -20˚C. Indirect flight muscles were dissected, washed and then fixed with 4% paraformaldehyde in relaxing solution (20 mM Na-Phosphate pH 7.2, 2 mM MgCl$_2$, 2 mM EGTA, 5 mM DTT, 5 mM ATP) with protease inhibitors. The incubation of rat anti-α-actinin antibody MAC276 (1:100; Babraham Bioscience Technologies) together with Alexa 594-phalloidin (1:100; ThermoFisher Scientific) was carried out overnight at 4˚C, followed by secondary antibody incubation for 3 hours at room temperature. Fluorescently labeled secondary antibody of the Alexa series (ThermoFisher Scientific) was used at a 1:400 dilution. Samples were mounted in ProLong Gold antifade solution (ThermoFisher Scientific). All images were acquired using a 63x 1.4 NA HC Plan Apochromat oil objective on a Leica SP8 confocal microscope. One-way ANOVA followed by Tukey's multiple mean difference post hoc tests were performed to determine statistically significant differences between genotypes using Prism 7 software (GraphPad).

### Supporting information

**S1 Fig. Purification of GST-Zasp52 fusion proteins.** Coomassie staining of Zasp52 domain GST fusions run on a SDS-PAGE gel after purification. (**A**) GST, Zasp52-PK and Zasp52-LIM234. Asterisks indicate fusion proteins. (**B**) GST, Zasp52-PP, Zasp52-PPG25W, and Zasp52-PPG26D PDZ domain mutants. Asterisk indicates fusion proteins. (**C**) GST, Zasp52-PK and Zasp52-PKP18DW19F PDZ domain mutant. (**D**) GST, human ZASP-PDZ, Zasp52-STOP143, human PDLIM7-PDZ, human ALP-PDZ, and human PDLIM2-PDZ. Molecular weight marker is indicated in kD.
(TIF)

**S2 Fig. Entire blot of Fig 3A showing actin.**
(TIF)

**S3 Fig. Entire blot of Fig 4A and 4B showing a Coomassie staining of actin and Zasp52.**
(TIF)

**S4 Fig. Entire blot of Fig 4C showing actin.**
(TIF)

**S5 Fig. Entire blot of Fig 4D showing actin.**
(TIF)

**S6 Fig. Entire blot of Fig 4E showing actin.**
(TIF)

**S7 Fig.**
(TIF)

**S8 Fig.**
(TIF)

## Acknowledgments

We thank the CIAN imaging facility for help with confocal microscopy.

## Author Contributions

**Conceptualization:** Kuo An Liao, Frieder Schöck.

**Data curation:** Nicanor González-Morales, Frieder Schöck.

**Formal analysis:** Nicanor González-Morales.

**Funding acquisition:** Frieder Schöck.

**Investigation:** Kuo An Liao, Nicanor González-Morales.

**Methodology:** Kuo An Liao, Nicanor González-Morales.

**Project administration:** Frieder Schöck.

**Supervision:** Frieder Schöck.

**Visualization:** Frieder Schöck.

**Writing – original draft:** Kuo An Liao, Frieder Schöck.

**Writing – review & editing:** Frieder Schöck.

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
