## [Decision Letter · Decision Letter 0]

17 Mar 2020

PONE-D-20-02049

Characterizing the actin-binding ability of Zasp52 and its contribution to myofibril assembly

PLOS ONE

Dear Dr. Schöck,

Thank you for submitting your manuscript to PLOS ONE. After careful consideration, we feel that it has merit but does not fully meet PLOS ONE’s publication criteria as it currently stands. Therefore, we invite you to submit a revised version of the manuscript that addresses the points raised during the review process.

Please respond to comments raised by Reviewers.

Regarding blot/gel data: PLOS ONE now requires that submissions reporting blots or gels include original uncropped blot/gel image data as a supplement or in a public repository. This is in addition to complying with our image preparation guidelines described at https://journals.plos.org/plosone/s/figures#loc-blot-and-gel-reporting-requirements. These requirements apply both to the main figures and to cropped blot/gel images included in Supporting Information.

We would appreciate receiving your revised manuscript by May 01 2020 11:59PM. To enhance the reproducibility of your results, we recommend that if applicable you deposit your laboratory protocols in protocols.io, where a protocol can be assigned its own identifier (DOI) such that it can be cited independently in the future. For instructions see: http://journals.plos.org/plosone/s/submission-guidelines#loc-laboratory-protocols

We look forward to receiving your revised manuscript.

Kind regards,

Sang-Chul Nam, Ph.D.

Academic Editor

PLOS ONE

Journal Requirements:

Reviewers' comments:

Reviewer's Responses to Questions

**Comments to the Author**

1. Is the manuscript technically sound, and do the data support the conclusions?

Reviewer #1: Yes

Reviewer #2: Yes

Reviewer #3: Yes

2. Has the statistical analysis been performed appropriately and rigorously? 

Reviewer #1: Yes

Reviewer #2: Yes

Reviewer #3: Yes

3. Have the authors made all data underlying the findings in their manuscript fully available?

Reviewer #1: Yes

Reviewer #2: Yes

Reviewer #3: Yes

4. Is the manuscript presented in an intelligible fashion and written in standard English?

Reviewer #1: Yes

Reviewer #2: Yes

Reviewer #3: Yes

5. Review Comments to the Author

Reviewer #1: This manuscript by Liao et al. dissected the role of Zasp52 in sarcomere formation in Drosophila. Zasp52 is a member of Zasp PDZ domain family in Drosophila and is characterized as a Z-disc protein. While previous studies demonstrated that Zasp52 interacts with α-actinin to regulate myofibril assembly, the mechanism through which Zasp 52 co-ordinates muscle development remains unknown. Here the authors have shown that Zasp52 regulates myofibrillogenesis through its interaction with actin filaments. Furthermore, the extended PDZ domain of Zasp52 protein is critical for its binding with actin during myofibril assembly. Finally, the authors demonstrated that the α -actinin-binding domain of Zasp52 is not sufficient to rescue myofibril associated defects in Zasp52 mutants. These studies further highlight the novel role of actin binding domain of Zasp52 during indirect flight muscle assembly.

The paper is well written and is appropriate for PLOS One. However, there are some points that authors should address.

1. Authors should clearly explain box plot quantification of sarcomere defects associated with different genotypes in figure 1.

2. In figure 2 A, authors should explain the reason for the presence of a faint band in lane 2 of the blot containing Mef2-Gal4 control extracts.

3. In figure 2 B, authors have performed GST pull down experiments to show the interaction between actin and Zasp52. They should specify the sample set that was used as an input. Also, these panels should also include anti-zasp52 or anti-flag stains to confirm that the pull-down was successful.

4. In figure 4 the authors mentioned that Zasp52-STOP143 partially rescued Zasp52MI02988/Df myofibril defects, but the representative image does not show any rescue of sarcomeric defects compared to Zasp52MI02988/Df alone.

5. In figure 5, authors should explain how the Actin88F mutant enhances the pupal lethality of Zasp 66 knock down flies, since Act88F is thought to be expressed only in the flight muscles, and mutants are homozygous viable. Can this lethality be rescued by expression of wild-type Actin88F?

Reviewer #2: In this manuscript Liao et al. begun to characterize the actin binding ability of Zasp52, a Drosophila member of the Alp/Enigma protein family. They found that besides �-actinin binding, the N-terminal PDZ domain and ZM region of the protein confers actin binding, and present genetic data suggesting that �-actinin and actin binding are both required for full rescue of the myofibrillar defects exhibited by Zasp52 mutants. This is a compact, well-written paper dealing with a potentially very interesting, novel function of this highly conserved protein family thought to be critical for Z-disk organization and maintenance. The experimental approach involved the combination of some standard techniques of fly genetics and biochemistry; the experiments were clearly described and reasonably interpreted. I believe that the dataset gained is valuable for investigators of this field by further confirming an important new aspect of the scaffolding role of these Z-disk proteins. Nevertheless, the manuscript could be improved by clarifying a couple of questions listed in my major and minor comments.

Major points:

1. One major finding of the paper is that in vitro Zasp52 is able to interact both with G-actin and F-actin. In all likelihood, the in vivo relevance of G-actin binding is negligible (or not trivial, at best), by contrast, F-actin binding can indeed be important for myofibril assembly and Z-disk formation. Unfortunately, most data presented are about analysis of the Zasp52/G-actin interaction which may not be the same and relevant as to F-actin interaction. Thus, it would be important to characterize the Zasp52/F-actin interaction with the same truncated and mutant versions as it was done for Zasp52/G-actin.

2. The genetic interaction data presented in Figure 5. are somewhat surprising. Heterozygosity for the muscle specific Act88F null mutation is not known to affect viability, and even the homozygous null mutants are viable. Therefore, I wonder about the relevance and reliability of the results revealing an effect on the ratio of pupal lethality. In addition, I lack the evidence for the interaction in muscles, in particular, in the IFM. Muscle phenotypes could largely enhance this line of experiments. A minor issue (or more likely a mistake by the authors) is that in Figure 5. B, C the lethality shown for Act88F/+ is zero % (lower than for wild type) and no error bar is shown.

Minor points:

1. In Figure 2. A an anti-Flag Western blot would help to assess the efficiency of the immunoprecipitation.

2. In Figure 2. B Zasp52LIM234 does not appear to bind G-actin in this GST pull-down assay, however, according to Figure S1A, this protein is expressed/bound to the beads in a much lower level as compared to Zasp52PK. Can the authors comment on this point? Did they make any efforts to work with equal protein amounts in the pull-down assays?

3. Results of this paper suggest that, although with significantly different affinities, Zasp52 is capable of binding to �-actinin as well as to that of actin. This is an exciting result, however, it remained unclear how it mechanistically comes together at the Z-disk. Is it a competitive or allosteric binding with regard to actin? It would be interesting to see experiments where all three proteins (Zasp52, �-actinin and F-actin) are present, and ideally, it would be very useful to create Zasp52 mutations that separate �-actinin binding from actin binding. In the absence of those, the authors could speculate in the Discussion how they think about this issue.

Reviewer #3: The manuscript from Liao et al builds upon previous studies of the Schock lab, which helped establish and elucidate the roles of Z-disc associated Alp/Enigma family proteins, and in particular Zasp52, in assembly of Drosophila indirect flight muscle (IFM) myofibrils and sarcomeres. The focus of the current study is on the functional significance of the actin-binding capacity of Zasp52, and on characterization of molecular details of the Zasp52-actin interaction.

The capacity of mammalian Alp/Enigma family proteins to bind/associate with both monomeric and filamentous actin has been previously demonstrated and the molecular details of these interactions have been fleshed out. The findings reported in the current study contribute to the field in two major ways:

• Genetic interactions suggest that the actin-binding capacity of Zasp52 is important for proper myofibril/sarcomere assembly in an in vivo setting;

• At the molecular level, the PDZ domains of Zasp52 and other Alp/Enigma family proteins, which constitute the link to a-actinin and the Z disc, are shown to contribute to their actin-binding abilities as well.

While these are rather modest advances, there is no question that the experimental work that backs them up is convincing and of good quality (see, however, some comments below), and furthermore, they are likely to encourage additional work on the significance of the Zasp-actin interaction and the molecular mechanisms involved, topics that appear to deserve considerable more attention.

Specific comments on the manuscript:

• P5 line 107: Actin88F is said to be the only actin gene expressed in IFMs. While this may be the case, this issue has not been fully resolved in the literature, and the evidence put forward in the cited reference (Nongthomba et al) is indirect. I suggest a less definitive statement such as: “Act88F is one of six actin-coding genes in Drosophila, and is the major and possibly the only one expressed in indirect flight muscles”.

• P5 line 111/Figure 1E: While clearly significant, the quantitative differences in myofibril/sarcomeric structure between Act88F heterozygotes and Zasp52/Act88F trans-heterozygotes seem less pronounced then might be expected from the images shown. Has the flight capacity of these flies been examined?

• P6 line 139/Figure 2A: The text states that extracts from the control Mef2-GAL4 line “do not” show an interaction with G-actin in the pull-down/incubation assay, when in fact, they weakly do (as correctly stated in the figure legend)- the main text should be modified.

• Figure 2- it is essential that a schematic representation of the different Zasp52 domains and constructs used in this study (similar to the scheme in Fig S1) appear in a main text figure such as figure 2, to allow readers a readily available resource for understanding the different experiments and following the arguments made.

• P7 line 170: To avoid ambiguity replace His-Zasp52 with His-Zasp52-FLAG.

• P8 lines 210-211: I think that a more accurate and thought-provoking way of summarizing the GST pull-down assays for identifying actin-binding domains is: “These data imply that both the ZM and PDZ domains are essential contributors to the actin-binding capacity of Zasp52, suggesting a more complex molecular basis for this interaction than previously recognized”.

• P9 line 232: Insert “binding alone” so that the sentence reads “indicating that a-actinin binding alone is not sufficient…

• Figure 4 title/line 234-5. The title is inappropriate as it represents a conclusion the authors derive from the data rather than a description of what the figure shows. “Limited rescue ability of a Zasp52 form defective in actin binding” is preferable.

• Figure 5 and related text. In my opinion, this final section of the experimental results adds little to the study. The conclusion of a “biochemical interaction” between Zasp66 and actin is based on a very indirect phenotypic assay (pupal lethality), and no molecular data is provided to back up. It is my strong recommendation that the entire section be taken out.

6. PLOS authors have the option to publish the peer review history of their article (what does this mean?). If published, this will include your full peer review and any attached files.

Reviewer #1: No

Reviewer #2: No

Reviewer #3: No

---

## [Author Response · Author response to Decision Letter 0]

6 Apr 2020

Response to Reviewers

We thank the reviewers for their thoughtful response. Below we provide a detailed response to all the points raised by the reviewers. We tried our best to answer all the reviewers' comments, despite McGill University being closed for an indeterminate length of time due to Covid-19.

As requested by PLOS ONE policy, we added the entire blots except for figure 4C (formerly 3C). The first author has left the lab 2 years ago, and despite diligently searching the hard drive he left behind, we could not find this blot. However, all these blots are streptavidin detections of biotinylated actin (figure 4C, D, E, F and 3A, B), so they all look quite similar. Furthermore, the data of figure 4C is confirmed by figure 4D, E, and F. In particular, figure 4E makes a very strong point that the PDZ domain is essential for actin binding. There we mutated 2 amino acids in the 450 amino acid long Zasp52-PK protein (these 2 aa are within the 90 aa PDZ domain of Zasp52-PK). This largely disrupts actin binding. Fig. S1D additionally shows that we purified wild type and mutant Zasp52-PK similarly well, indicating that these mutations did not affect the stability of Zasp52. Therefore it is reasonable to conclude that the PDZ domain plays an important role in actin binding.

Reviewer 1:

1. Authors should clearly explain box plot quantification of sarcomere defects associated with different genotypes in figure 1.

We counted the number of sarcomeres present in each image for each of our four conditions. All the images were taken with comparable parameters. Because the indirect flight muscles are very regular, they have similar sarcomere numbers. In mutant indirect flight muscles, many sarcomeres are missing resulting in lower sarcomere numbers. This quantitative approach has been previously reported (Liao KA et al., 2016).

2. In figure 2 A, authors should explain the reason for the presence of a faint band in lane 2 of the blot containing Mef2-Gal4 control extracts.

When using protein extracts from fly thorax (1000s of proteins), it is normal to observe background binding to Flag M2 beads, to which in turn some actin binds. 

We added the Flag blot to Supplemental Figures showing the entire blots.

3. In figure 2 B, authors have performed GST pull down experiments to show the interaction between actin and Zasp52. They should specify the sample set that was used as an input. Also, these panels should also include anti-zasp52 or anti-flag stains to confirm that the pull-down was successful.

The input is G-actin on its own. The presence of actin in lane 3 demonstrates the success of the pull-down. The success of protein purification of GST-tagged proteins is shown in Figure S1A.

4. In figure 4 the authors mentioned that Zasp52-STOP143 partially rescued

Zasp52MI02988/Df myofibril defects, but the representative image does not show any rescue of sarcomeric defects compared to Zasp52MI02988/Df alone.

Figure 5D (formerly 4D) shows a partial rescue, because Z-discs are always visible, even though myofibrils are smaller and Z-discs are partially disrupted. In contrast, in Figure 5B, there are entire sarcomeres without any discernible Z-discs, a much stronger phenotype. In order not to rely on individual images, we have quantified the rescue in figure 5E.

5. In figure 5, authors should explain how the Actin88F mutant enhances the pupal lethality of Zasp 66 knock down flies, since Act88F is thought to be expressed only in the flight muscles, and mutants are homozygous viable. Can this lethality be rescued by expression of wild-type Actin88F?

We have deleted figure 5.

Reviewer 2:

1. One major finding of the paper is that in vitro Zasp52 is able to interact both with G-actin and F-actin. In all likelihood, the in vivo relevance of G-actin binding is negligible (or not trivial, at best), by contrast, F-actin binding can indeed be important for myofibril assembly and Z-disk formation. Unfortunately, most data presented are about analysis of the Zasp52/G-actin interaction which may not be the same and relevant as to F-actin interaction. Thus, it would be important to characterize the Zasp52/F-actin interaction with the same truncated and mutant versions as it was done for Zasp52/G-actin.

Yes, it is true that most data are on the interaction with G-actin. We had to start somewhere, and F-actin interaction assays are considerably more difficult. We certainly plan to characterize the F-actin interaction in much more detail in the future.

2. The genetic interaction data presented in Figure 5. are somewhat surprising. Heterozygosity for the muscle specific Act88F null mutation is not known to affect viability, and even the homozygous null mutants are viable. Therefore, I wonder about the relevance and reliability of the results revealing an effect on the ratio of pupal lethality. In addition, I lack the evidence for the interaction in muscles, in particular, in the IFM. Muscle phenotypes could largely enhance this line of experiments. A minor issue (or more likely a mistake by the authors) is that in Figure 5. B, C the lethality shown for Act88F/+ is zero % (lower than for wild type) and no error bar is shown.

We deleted figure 5.

1. In Figure 2. A an anti-Flag Western blot would help to assess the efficiency of the

immunoprecipitation.

We added the anti-Flag Western blot to Supplemental Figures showing the entire blots.

2. In Figure 2. B Zasp52LIM234 does not appear to bind G-actin in this GST pull-down assay, however, according to Figure S1A, this protein is expressed/bound to the beads in a much lower level as compared to Zasp52PK. Can the authors comment on this point? Did they make any efforts to work with equal protein amounts in the pull-down assays?

LIM domains are very difficult to purify, which explains why we could only obtain smaller amounts. Yes, we worked with equal protein amounts. It should be kept in mind that pull-downs are qualitative assays. As we saw no binding at all, it seems reasonable to conclude that LIM domains of Zasp52 play no role in actin binding. However, this being a negative result, we cannot be certain.

3. Results of this paper suggest that, although with significantly different affinities, Zasp52 is capable of binding to -actinin as well as to that of actin. This is an exciting result, however, it remained unclear how it mechanistically comes together at the Z-disk. Is it a competitive or allosteric binding with regard to actin? It would be interesting to see experiments where all three proteins (Zasp52, -actinin and F-actin) are present, and ideally, it would be very useful to create Zasp52 mutations that separate -actinin binding from actin binding. In the absence of those, the authors could speculate in the Discussion how they think about this issue.

We added a sentence to the discussion on the questions raised by this work. Certainly, we plan to identify amino acids that uniquely disrupt actinin versus F-actin binding in the future, but we have not been successful so far.

Reviewer 3:

• P5 line 107: Actin88F is said to be the only actin gene expressed in IFMs. While this may be the case, this issue has not been fully resolved in the literature, and the evidence put forward in the cited reference (Nongthomba et al) is indirect. I suggest a less definitive statement such as: “Act88F is one of six actin-coding genes in Drosophila, and is the major and possibly the only one expressed in indirect flight muscles”.

Thanks for pointing this out. We changed the statement.

• P5 line 111/Figure 1E: While clearly significant, the quantitative differences in

myofibril/sarcomeric structure between Act88F heterozygotes and Zasp52/Act88F transheterozygotes seem less pronounced than might be expected from the images shown. Has the flight capacity of these flies been examined?

Yes, we tested flight capacity: wild type and Zasp52/+ can fly normally, Act88F/+ and Act88F/Zasp52 cannot fly at all. As this adds no insight, we did not show the data.

• P6 line 139/Figure 2A: The text states that extracts from the control Mef2-GAL4 line “do not” show an interaction with G-actin in the pull-down/incubation assay, when in fact, they weakly do (as correctly stated in the figure legend)- the main text should be modified.

We modified the main text and figure legend.

• Figure 2- it is essential that a schematic representation of the different Zasp52 domains and constructs used in this study (similar to the scheme in Fig S1) appear in a main text figure such as figure 2, to allow readers a readily available resource for understanding the different experiments and following the arguments made.

We added a new figure 2 showing schematics of all constructs used.

• P7 line 170: To avoid ambiguity replace His-Zasp52 with His-Zasp52-FLAG.

Done

• P8 lines 210-211: I think that a more accurate and thought-provoking way of summarizing the GST pull-down assays for identifying actin-binding domains is: “These data imply that both the ZM and PDZ domains are essential contributors to the actin-binding capacity of Zasp52, suggesting a more complex molecular basis for this interaction than previously recognized”.

Changed

• P9 line 232: Insert “binding alone” so that the sentence reads “indicating that a-actinin binding alone is not sufficient…

Done 

• Figure 4 title/line 234-5. The title is inappropriate as it represents a conclusion the authors derive from the data rather than a description of what the figure shows. “Limited rescue ability of a Zasp52 form defective in actin binding” is preferable.

Changed

• Figure 5 and related text. In my opinion, this final section of the experimental results adds little to the study. The conclusion of a “biochemical interaction” between Zasp66 and actin is based on a very indirect phenotypic assay (pupal lethality), and no molecular data is provided to back up. It is my strong recommendation that the entire section be taken out.

We deleted figure 5.

---

## [Editor Report · Decision Letter 1]

8 Apr 2020

Characterizing the actin-binding ability of Zasp52 and its contribution to myofibril assembly

PONE-D-20-02049R1

Dear Dr. Schöck,

We are pleased to inform you that your manuscript has been judged scientifically suitable for publication and will be formally accepted for publication once it complies with all outstanding technical requirements.

With kind regards,

Sang-Chul Nam, Ph.D.

Academic Editor

PLOS ONE
---

## [Editor Report · Acceptance letter]

22 Jun 2020

PONE-D-20-02049R1 

Characterizing the actin-binding ability of Zasp52 and its contribution to myofibril assembly 

Dear Dr. Schöck:

I'm pleased to inform you that your manuscript has been deemed suitable for publication in PLOS ONE. Congratulations! Your manuscript is now with our production department. 

Kind regards, 

on behalf of

Dr. Sang-Chul Nam 

Academic Editor

PLOS ONE